# Building Corner Detection in Aerial Images with Fully Convolutional Networks

**DOI:** 10.3390/s19081915

**Published:** 2019-04-23

**Authors:** Weigang Song, Baojiang Zhong, Xun Sun

**Affiliations:** 1School of Computer Science and Technology, Soochow University, Suzhou 215006, China; m1273152693h@163.com (W.S.); 20164027009@stu.suda.edu.cn (X.S.); 2Provincial Key Laboratory for Computer Information Processing Technology, Soochow University, Suzhou 215006, China

**Keywords:** building, corner detection, convolutional networks, semantic segmentation, aerial image, conditional random fields

## Abstract

In aerial images, corner points can be detected to describe the structural information of buildings for city modeling, geo-localization, and so on. For this specific vision task, the existing generic corner detectors perform poorly, as they are incapable of distinguishing corner points on buildings from those on other objects such as trees and shadows. Recently, fully convolutional networks (FCNs) have been developed for semantic image segmentation that are able to recognize a designated kind of object through a training process with a manually labeled dataset. Motivated by this achievement, an FCN-based approach is proposed in the present work to detect building corners in aerial images. First, a DeepLab model comprised of improved FCNs and fully-connected conditional random fields (CRFs) is trained end-to-end for building region segmentation. The segmentation is then further improved by using a morphological opening operation to increase its accuracy. Corner points are finally detected on the contour curves of building regions by using a scale-space detector. Experimental results show that the proposed building corner detection approach achieves an F-measure of 0.83 in the test image set and outperforms a number of state-of-the-art corner detectors by a large margin.

## 1. Introduction

Building corners, one of the most useful types of geometric information, can effectively describe the structure of buildings in urban areas. They have therefore been widely exploited to solve a number of tasks including city modeling [1] and geo-localization [2,3], among many others. In this paper, the problem of building corner detection in aerial images is investigated and an efficient approach is developed to solve it.

Over the past decades, a number of generic corner detectors have been proposed, which can be broadly classified into three groups as follows: intensity-based algorithms [4], contour-based algorithms [5,6,7], and model-based algorithms [8]. The intensity-based algorithms detect corner points by checking the change of local gray values in the input image. The contour-based detectors, such as the curvature scale-space (CSS) algorithm [5], first extract contour curves with an edge detector and then obtain corner points by searching for curvature maxima along those curves. The model-based detectors find corners by matching image patches to a set of predefined corner models.

The existing generic detectors are designed to identify corners from normal images. However, in some kinds of vision tasks, the desired corner points could be defined with task-specific constraints. In particular, it is observed that the generic detectors are incapable of detecting building corners from aerial images. The reason is that there exist many disturbances in aerial images, e.g., corners on trees, shadows, and road signs. As a result, many false positives could be produced, as demonstrated in Figure 1. In this work, we will incorporate the concept of semantic image segmentation into the corner detection process to overcome this difficulty. It will be shown that building corners can be reliably detected while non-building ones are suppressed. The proposed approach could also be extensively used, with some changes, to solve other specified corner detection tasks.

The goal of semantic image segmentation is to assign a category label to each pixel in the input image, that is, to conduct a pixel-level classification of image regions. In recent years, AlexNet [10], the VGG [11], GoogLeNet [12], and ResNet [13] have significantly promoted the development of image classification. Based on these achievements, rapid progress in image semantic segmentation has been inspired by fully convolutional networks (FCNs) [14], in which the last fully-connected layers of previous well-known classification networks are replaced with convolutional ones.

Although FCN semantic image segmentation delivers a superior performance compared to traditional methods, there are still several problems. First, its consecutive pooling operations could abandon pixel-level spatial information and reduce the resolution of image features. For that, a dilated convolution can be used so that the networks may control the resolution of feature maps [15,16]. Second, objects at multiple scales require an integration of information from various spatial scales. One effective approach is to employ the strategy of multi-scale feature fusion. In [17], it is suggested to exploit multi-scale inputs by resizing the image at several different scales and then fuse the features produced from all of the scales. DeepLab [15] uses atrous spatial pyramid pooling, (ASPP), where side-by-side dilated convolution layers with different rates are used to obtain multi-scale context information. Another method is to apply a post-processing stage using a conditional random field (CRF) [15,18]. In this work, we employ DeepLab for building region segmentation since it combines most of the above-mentioned merits.

The remainder of the paper is organized as follows. DeepLab for semantic image segmentation is briefly reviewed in Section 2. Our new algorithm for building corner detection is proposed and investigated in detail in Section 3. Experimental results and discussions are presented in Section 4. Finally, conclusions are drawn in Section 5.

## 2. Semantic Image Segmentation with the DeepLab

DeepLab [15] is comprised of two well-known modules, i.e., FCNs and CRFs. For an input image, its semantic segmentation result is produced by the FCNs, and this is followed by the use of the CRFs to achieve higher segmentation accuracy.

### 2.1. Atrous Convolution

FCNs have been successfully used for semantic image segmentation by deploying networks in a fully convolutional fashion. To deal with the down-sampling issue of the input image during the convolution operations in the FCNs, atrous convolution (or dilated convolution) [19] is employed, by which a given resolution of feature maps can be yielded so that the field-of-view of filters are enlarged. For a one-dimensional (1-D) input *x* with a filter *w* of length *K*, the atrous convolution is defined as follows:(1)$$y\left[i\right]=\sum_{k=1}^{K}x\left[i+r\cdot k\right]w\left[k\right],$$
where *r* is a rate parameter representing the stride with which the input signal is sampled and $y\left[i\right]$ denotes the output. When r=1, the atrous convolution degenerates to a standard convolution operation. For a two-dimensional (2-D) input, the implementation of the atrous convolution operation is similar to the 1-D case.

Due to a trade-off between efficiency and accuracy, atrous convolution is used in a chain of layers to increase the final resolution from 1/32 to 1/8 of the original image.

### 2.2. Multiscale Processing

To enhance the FCNs for capturing both local and global information, multi-scale feature fusion is used [17]. To be specific, several parallel FCNs (three in this work) are first implemented with the same parameters to extract feature maps at multiple scales for an input image, and then these feature maps are fused to improve the segmentation performance. In addition, several parallel atrous convolutional layers with diverse sampling rates are employed to extract context information in different ranges. This technique is called atrous spatial pyramid pooling (ASPP).

### 2.3. Fully Connected Conditional Random Fields

FCNs generate a semantic segmentation result; however, the result is often inaccurate and therefore cannot precisely predict the borders of objects in the segmentation map. To improve the segmentation accuracy, the fully connected CRFs model is exploited. Define $\theta_{i}=-\log P\left(x_{i}\right)$, where $P\left(x_{i}\right)$ is the label allocation probability at the $i^{\mathrm{th}}$ pixel computed by the FCNs. Let $\theta_{ij}$ represent a quantized relationship between two pixels defined as
(2)$$\begin{array}{c} \theta_{ij}=\mu \left\{w_{1}\exp\left(-\frac{{∥p_{i}-p_{j}∥}^{2}}{2\sigma_{\alpha }^{2}}-\frac{{∥I_{i}-I_{j}∥}^{2}}{2\sigma_{\beta }^{2}}\right)+w_{2}\exp\left(-\frac{{∥p_{i}-p_{j}∥}^{2}}{2\sigma_{\gamma }^{2}}\right)\right\}, \end{array}$$
where μ equals 1 if $x_{i}\neq x_{j}$ and 0 otherwise. The two terms on the right-hand side of Equation (Equation 2) use two Gaussian kernels in different feature spaces, respectively. The first one takes into account pixel locations (denoted by *p* ) as well as RGB color (denoted by *I*), and the second one only considers pixel locations. The CRFs model applies an energy function in the form of
(3)$$E\left(x\right)=\sum_{i}\theta_{i}\left(x_{i}\right)+\sum_{ij}\theta_{ij}\left(x_{i},x_{j}\right),$$
where *x* is the label allocation for pixels.

By incorporating FCNs, multi-scale processing, and fully connected CRFs, DeepLab is able to achieve a satisfactory performance.

## 3. Proposed Algorithm for Building Corner Detection

Our new algorithm for building corner detection mainly contains three steps: building region segmentation, building contour extraction, and building corner detection, which will be developed step-by-step and described in detail.

### 3.1. Dataset of Aerial Images

The dataset of aerial images used for training (11,700 images of 321 × 321 pixels) and testing (450 images of 321 × 321 pixels) is from the Vaihingen dataset for the ISPRS 2D semantic labeling contest [20]. The Vaihingen dataset contains 33 true orthophoto tiles of high resolution with three channels (that is, near-infrared, red, and green), and each channel corresponds to a digital surface model (DSM) derived from dense image-matching techniques. It comprises abundant surface information with a spatial resolution of 9 cm; however, labeled ground truth is only available for some of the images (16 labeled images from a total of 33 images). Six surface categories have been annotated in the dataset, namely, *impervious surfaces*, *buildings*, *low vegetation*, *trees*, *cars*, and *background*.

In our work, the building regions are focused on, and therefore the surface categories defined in our dataset only include *buildings* and *background*. In addition, for a fast implementation of our developed algorithm, we only pay attention to the common aerial image data and abandon the additional DSM information. As a result, a new dataset, without DSM, is generated by marking out *building* areas from *background* only. Similar to [21], we further split these 16 newly-produced images with labeled ground truth into training (including #1, #3, #5, #7, #13, #15, #17, #21, #23, #26, #28, #32, and #34) and testing images (including #11, #30, and #37). We randomly sample 150 patches of 321 × 321 pixels from each training image, and we conduct data augmentation including image flipping and rotation of 90 degrees, 180 degrees and 270 degrees. In total, 11,700 new and smaller training images are therefore obtained to determine the FCN parameters which have been pre-trained on MS-COCO [22]. In the same way, 450 images are also produced for testing. Figure 2 shows two example images as well as the ground-truth segmentation results for our developed dataset.

### 3.2. Building Segmentation

DeepLab based on ResNet-101 [13] is used to generate the initial segmentation of building regions because of its superior performance. The ResNet-101 model contains a 7 × 7 convolutional layer and 5 residual blocks. Each residual block includes several 3 × 3 or 1 × 1 convolutional layers. The latter two blocks are re-purposed by atrous convolution, as described previously in Section 2.1. Pre-training is then conducted on MS-COCO [22]. When training with the 11,700 images in our dataset, 40 K iterations are implemented, where the batch size is taken to be 1. The base learning rate is set to $2.5\times 10^{-4}$ and a “poly” learning rate policy is employed. The values of momentum and weight decay are 0.9 and $5\times 10^{-4}$, respectively.

Figure 3 illustrates two examples of our testing. The first column of the figure presents the test images, and the second column shows the ground-truth segmentation. It can be seen from the third column that the FCN (DeepLab without CRF) can extract most building regions; however, the segmentation maps are somewhat blurred and inaccurate. By using the CRF for post-processing, the boundary edges of buildings become sharper, as shown in the fourth column. Unfortunately, some segmentation flaws such as spur-like artifacts are still observed, which could incur many false positives in the subsequent corner detection process.

The morphological opening operation, which works as a basic tool of morphological noise removal in computer vision and image processing, is exploited to remove the segmentation flaws produced by the FCN-CRF, as mentioned above. This operation is defined by
(4)$$A\circ B=\left(A⊖B\right)⊕B,$$
where *A* is the input image, *B* is a structural element, and ⊖ and ⊕ denote the morphological erosion and dilation operations, respectively.

For the experiment demonstrated in Figure 3, the results after using the morphological opening operation are shown in the last column of the figure. For convenience, the DeepLab model with a CRF post-processing step and the morphological opening operation (MOP) for flaw removal is denoted as FCN-CRF-MOP. For an objective evaluation, Table 1 shows a comparison of the above-mentioned segmentation methods in terms of the Intersection over Union (IoU) metric. One can see that our model (mean IoU = 93.21%) clearly outperforms the FCN (mean IoU = 92.83%) and FCN-CRF (mean IoU = 93.16%). This verifies that the flaw removal step using the opening operation can improve the accuracy of segmentation.

For the task of building corners detection, it is of crucial importance that the segmentation method used can produce enough accuracy along building boundaries. Therefore, the mean IoU within the trimap region (a narrow band along object boundaries, as defined in [23]) has also been exploited to evaluate different segmentation methods. For a demonstration, Figure 4 shows the trimaps on an example image with a width of 5 and 10 pixels. To evaluate our segmentation model, Table 2 documents the mean IoU measured within the trimaps of different widths on all 450 test images. It is observed that the measured accuracy of each compared method increases as the trimap width increases (that is, when a less-strict evaluation condition is imposed); however, our proposed model (i.e., the FCN-CRF-MOP) always outperforms the other two methods. This further verifies that the opening operation is able to improve the segmentation performance along building boundaries.

According to the objective evaluations of overall regional segmentation and marginal part segmentation, it can be concluded that our segmentation model can provide higher accuracy for segmentation than existing models.

### 3.3. Building Contour Extraction

The Matlab function bwboundaries is used to extract the contour curves from the previously generated maps of building area segmentation. The parameter *CONN* in this function, which specifies the connectivity to use when tracing parent and child boundaries, is taken to be 8. Since we only need peripheral contours of buildings, the parameter of the target type is chosen as “noholes”. For each contour curve, this function returns a set of point coordinates. The set of extracted building contours is denoted as $\left\{C_{i}|i=\left[1,N\right]\right\}$, where *N* is the number of curves and $C_{i}$ contains *m* points describing the $i^{\mathrm{th}}$ curve, in the form of $C_{i}=\left\{p_{j}:\left(x_{j},y_{j}\right)|j=1,2,\ldots ,m\right\}.$

For a visual evaluation, Figure 5 presents the contour extraction results of six input aerial images. For each image, the extracted curves are very close to the labeled ground-truth curves, where different curves are marked in different colors. One can see that the building boundaries have all been precisely predicted. This will be very beneficial for conducting corner detection in the subsequent step.

### 3.4. Building Corner Detection

In generic corner detection, the extraction of contour curves is generally implemented by using the Canny edge detector. However, to detect corners from aerial images, precise contour extraction plays a much more important role than in generic corner detection. In fact, as the contour curves of buildings have been extracted with high accuracy (as shown in Figure 5), the corner detection step becomes rather easy. In the following, scale-space corner detection is conducted to obtain a reliable detection performance.

Our corner detection step is similar to the corner detection process proposed in [24], but with a simpler implementation since the scale-space tree phrasing is omitted (this tree phrasing operation is used to cope with complex shapes; however, the contour curves of a building are, in general, simple). Let r(s)=(x(s),y(s)) be the building contour under consideration, where *s* is the arc-length parameter. Scale-space corner detection [24] is based on a scale-space representation of the contour curve r(s), which can be expressed as the solution of a heat equation, as follows:(5)$$\frac{\partial r}{\partial t}=\frac{1}{2}\frac{\partial^{2}r}{\partial s^{2}},$$
where *t* the time variable. In the digital case, a contour curve is represented by a set of samples $\left\{\left(x_{i},y_{i}\right)\;|\;i\in Z\right\}$ (*i* using modulo *n* in the case of a closed curve that has *n* points) that are equally spaced with a spacing of Δs=1. To conduct a scale-space corner detection, a curve evolution is performed to generate a scale-space representation of the curve:(6)$$x_{i,m+1}=0.25x_{i-1,m}+0.5x_{i,m}+0.25x_{i+1,m};$$
(7)$$y_{i,m+1}=0.25y_{i-1,m}+0.5y_{i,m}+0.25y_{i+1,m},$$
where m=0,1,2,⋯ serves as a scale parameter of the evolution procedure, and $\left\{\left(x_{i,0},y_{i,0}\right)\right\}=\left\{\left(x_{i},y_{i}\right)\right\}$. The weightage {0.25,0.5,0.25} used above is derived from a discretization of the heat Equation (Equation 5). Then, the curvature of the curve is computed by using a standard curvature measure for digital curves [24]. To be specific, at the *i*th point we have
(8)$$\kappa_{i,m}=\frac{\delta x_{i,m}\delta^{2}y_{i,m}-\delta^{2}x_{i,m}\delta y_{i,m}}{{\left\{{\left(\delta x_{i,m}\right)}^{2}+{\left(\delta y_{i,m}\right)}^{2}\right\}}^{3/2}},$$
where δ and $\delta^{2}$ denote the first-order and second-order central difference operators given by
$$\delta f_{i,m}=0.5\left(f_{i+1,m}-f_{i-1,m}\right);$$
$$\delta^{2}f_{i,m}=f_{i+1,m}-2f_{i,m}+f_{i-1,m}.$$

The curvature measure presented above is in compliance with the standard curvature expression in the continuous case. It is therefore more straightforward than the one proposed by Rattarangsi and Chin [25]. Finally, corner points are recognized on a coarse scale (m=400) by performing a non-maximum suppression on $\kappa_{i,m}$ and then tracked back to the finest scale (m=0) for location improvement.

By an incorporation of the corner detection step and the semantic segmentation, a full algorithm for detecting building corners from aerial images is developed, as demonstrated in Figure 6. The main steps of the proposed algorithm are outlined below.

**Step** **1.***Building region segmentation*. With the trained DeepLab model, a building region segmentation of the input image is conducted using the FCNs (a ResNet-101 re-purposed by atrous convolution), followed by the exploitation of a CRF step and the morphological opening operation (MOP) to improve segmentation accuracy.**Step** **2.***Building contour extraction*. The Matlab function bwboundaries is employed to extract the contour curves on the segmentation map of building regions.**Step** **3.***Building corner detection*. Based on a scale-space representation of each building boundary, corner points are recognized at a coarse scale and then tracked back to the finest scale to improve their locations.

## 4. Experimental Results and Discussions

Ten images are selected for testing from our developed dataset of aerial images, as shown in Figure 10a. These images contain rich building corner information, as well as a large number of disturbances generated by trees and shadows. For an objective evaluation, we manually annotated the ground-truth corners on each test image (highlighted by yellow squares, as shown in Figure 10b).

### 4.1. Evaluation Metrics

Three evaluation metrics, i.e., the precision, recall, and F-measure, are adopted to evaluate the performance of corner detection. The precision measures the ability of the used approach to retrieve corners that are relevant to the ground truth; the recall measures the relevant corners that are actually detected; and the F-measure is a consolidated metric based on the precision and the recall. These three evaluation metrics are computed as follows:(9)$$Precision=\frac{N_{r}}{N_{r}+N_{w}}=\frac{N_{r}}{N_{d}},$$
(10)$$Recall=\frac{N_{r}}{N_{r}+N_{m}}=\frac{N_{r}}{N_{g}},$$
(11)$$F_{\beta }=\frac{\left(1+\beta^{2}\right)Precision\times Recall}{\beta^{2}Precision+Recall},$$
where $N_{r}$ is the number of true corners that have been detected; $N_{w}$ is the number of false corners that have been detected; $N_{d}$ is the total number of all detected corners; $N_{m}$ is the number of mis-detected corners; and $N_{g}$ is the number of ground-truth corners. During evaluation, it is observed that the recall rate of a generic corner detector is always much higher than its precision rate. To achieve a fair comparison, we set β=2 so that the evaluation pays more attention to the recall rate.

### 4.2. Verification of the FCN-CRF-MOP Model

Our algorithm is developed upon an effective building segmentation approach, i.e., the FCN-CRF-MOP model as proposed and discussed in Section 3.2. This model is of crucial importance for the final building corner detection. For verification, corner detection is conducted on the segmentation results produced with the existing FCN and FCN-CRF models and our proposed FCN-CRF-MOP model individually. Figure 7 shows a performance comparison of the three segmentation models with respect to the precision, recall, and F-measure of the detected corners. While it has been shown in Section 3.2 that the FCN-CRF-MOP model improves the segmentation accuracy of building regions, one can see from Figure 7 that this model also helps to generate better corner detection performance when compared with the FCN and FCN-CRF models.

### 4.3. Verification of Our Corner Detection Scheme

Our proposed building corner detector is based on the building segmentation results produced with a DeepLab model, followed by conducting a scale-space corner detection on the building contours. We treat this as an effective scheme for solving the building corner detection task. For verification, a reference algorithm is developed, which skips over the building segmentation step and extracts building contours directly to conduct corner detection. For the training data set, we use the building regions defined by human annotation as described in Section 3.1 to generate binary edge maps. Subsequently, a DeepLab model is trained on the new data set by using the method specified in Section 3.2. However, no edge information is produced at the testing stage. In fact, the loss function of the DeepLab is the sum of cross-entropy terms for each spatial position in the Convolutional Neural Networks (CNN) output map, where all positions and labels are equally weighted in the overall loss function. When DeepLab is directly used for edge detection, a strong imbalance between edge pixels and background pixels could cause the edge information be totally neglected.

To overcome the difficulty mentioned above, the existing CNN models usually employ a weighted loss function to keep a balance between background and edges. The Richer Convolutional Features (RCF) [26], a typical network of edge detection, is therefore exploited as the basic network for building contour detection. In our training process, the resolution of every input image is 321 × 321, and all the other network parameter settings are taken to be the default values of the RCF. Since each edge map produced from the RCF is in a probability format, a standard non-maximum suppression (NMS) is used to transform these into a binary ones for conducting the follow-up corner detection step.

Figure 8 shows the performance of the RCF in building contour extraction. The test images are shown in the first column, and the ground-truth building contours are shown in the second column. The outcome of the RCF is presented in the third column, and the binary edge maps produced by using non-maximum suppression is demonstrated in the fourth column. In the last column, the final results of curve extraction are presented, which are generated from the binary edge maps with several necessary operations such as redundant edge removal and adjacent edge merging. One can see that the reference algorithm is incapable of predicting building contours with high accuracy. To be specific, the predicted edges could be blurred, fragmented, or mis-detected.

Through a comparison of the performance of the reference contour extraction approach (as shown in the last column of Figure 8 and that of our approach (as shown in the last column of Figure 5, it can be concluded that the corner detection scheme proposed in this work represents a more effective way of detecting building corners from aerial images.

### 4.4. Objective Evaluation

The performance of our proposed corner detection algorithm is evaluated and compared with that of seven state-of-the-art corner detectors, including the ANDD [8], CSS [5], CPDA [27], Fast CPDA [28], He & Yung [29], GCM [30], and WEAE [9]. Specifically, a reference corner detector, i.e., corner detection based on the reference contour extraction approach as developed in Section 4.3, is also included for a comparison with the proposed corner detector. Both detectors share the same corner detection step as specified in Section 3.4, and a comparison between them will further verify the validity of our corner detection scheme developed upon the FCNs.

For each of the existing corner detectors under comparison, its parameters are optimized individually to get the best detection performance, as summarized in Table 3, where the parameter η denotes the threshold of the discrete curvature individually defined and exploited in every corner detector for suppressing redundant corner points (those generated from insignificant image structures or caused by images noise), and the parameter $T=(T_{l},T_{h})$ specifies the sensitivity threshold of the Canny edge detector used in building contour extraction. For our detector, the default parameters of DeepLab suggested in [15] are used to produce the building segmentation, and the structuring element of the morphological opening operation is chosen to be a disk with a radius of 3.

Figure 9 shows the evaluation results. It is observed that the existing corner detectors all have rather high recall rates, indicating that they are able to detect most of the building corner points. Unfortunately, their precision rates are all remarkably low, indicating that a lot of false positives are generated. As a result, they all have low F-measure values. Our proposed detector generates much fewer false positives and more true positives simultaneously. In consequence, it delivers the best results in terms of all of the evaluation metrics and outperforms the existing detectors, as well as the reference one, by a large margin.

### 4.5. Subjective Evaluation

For a subjective evaluation, Figure 10 shows a visual comparison of all of the evaluated corner detectors. The proposed detector produces the greatest number of true positives and shows strong robustness against disturbances, i.e., corners on other objects such as trees, building shadows, and roads. More specifically, for the first two test images, all building corner points were detected, as shown in the first and second rows. In the other cases as shown in the third to fifth rows, only a few building corner points are mis-detected. Although the other corner detectors under comparison are also able to detect most building corners, they yield many unwanted corners in non-building areas. These observations clearly indicate that the proposed corner detector is more suitable for detecting building corners than the existing detectors and the reference one.

### 4.6. Computational Complexity

The FCNs model for building segmentation is trained on a GPU (NVIDIA TITAN X), where the iteration number is taken to be 40,000. This training stage takes about 48 h. In Table 4, the run-times of different corner detectors are compared. For our new detector, the inference time of DeepLab for every aerial image is about 0.45 s, and the subsequent operation to improve its accuracy takes about 0.07 s. Therefore, the overall running time of our algorithm is 0.52 s per image, on average.

### 4.7. Discussion

The performance of our proposed corner detector relies on the accuracy of the extracted building contours. In most scenarios, the contours of buildings can be precisely predicted by using the FCN-CRF-MOP model developed in Section 3.2, and hence building corners are well detected. On the other hand, it is occasionally observed that for buildings with a complex structure, the extracted contours might be blurred (e.g., the third row of Figure 5c). In such a case, a sharp corner can become rounded and tends to be mis-detected (see the last low of Figure 10k). To cope with this problem, our future work will investigate in how to preserve the curvature of building contours. A possible way is to add the curvature values of each labeled building contour to the FCNs so that an edge-preserving network model can be trained for building contour extraction.

## 5. Conclusions

In this paper, we proposed an algorithm for detecting building corners in aerial images. The novelty of our work lies in the fact that an image’s semantic information is incorporated into the corner detection process by training a DeepLab network. To produce building segmentation results with high accuracy, a morphological opening operation is used to improve the performance of the network. Building corners are then reliably detected based on a scale-space representation of building contours. Both objective and subjective evaluations are conducted, and the results indicate that our proposed approach outperforms the state-of-the-art corner detectors by a large margin. To be specific, it achieves an F-measure of 0.83 on the test image set.

## Figures and Tables

**Figure 1 sensors-19-01915-f001:**
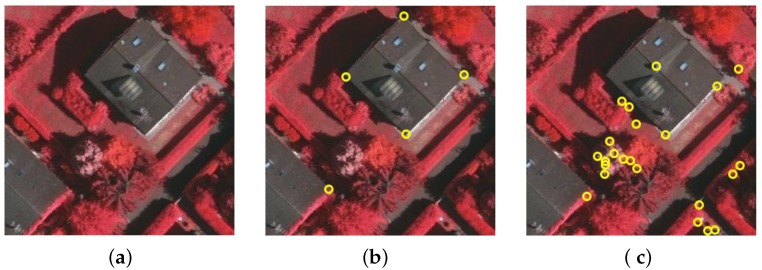
The problem to be solved in the present work: building corners in aerial images are hard to identify with existing generic corner detectors. (**a**) An aerial image for testing. (**b**) The ground-truth corner points (manually labeled). (**c**) Performance of a recently-proposed corner detection algorithm, the WEAE [9].

**Figure 2 sensors-19-01915-f002:**
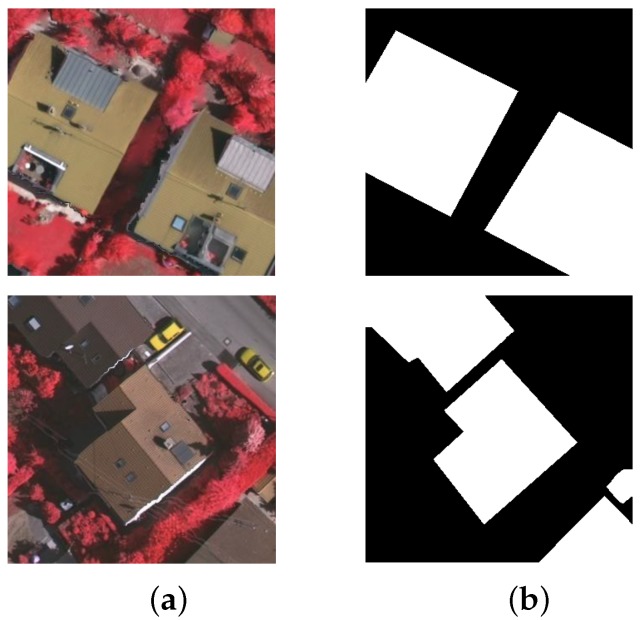
Two example images and the ground-truth segmentation results for our dataset: (**a**) Test images; (**b**) ground-truth building regions by human annotation.

**Figure 3 sensors-19-01915-f003:**
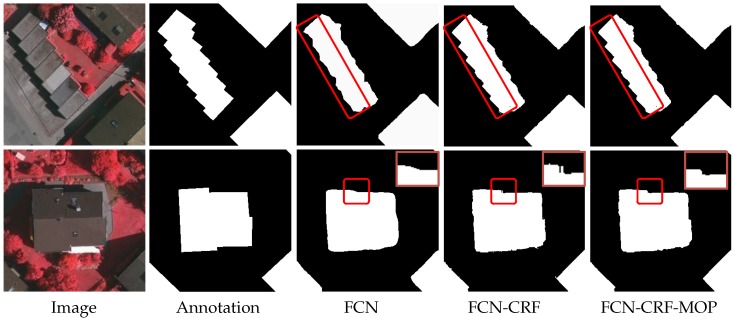
Segmentation results of the Fully Convolutional Network (FCN), the FCN with fully-connected Conditional Random Fields (FCN-CRF), and the DeepLab model with a CRF post-processing step and the morphological opening operation for flaw removal, denoted as FCN-CRF-MOP.

**Figure 4 sensors-19-01915-f004:**
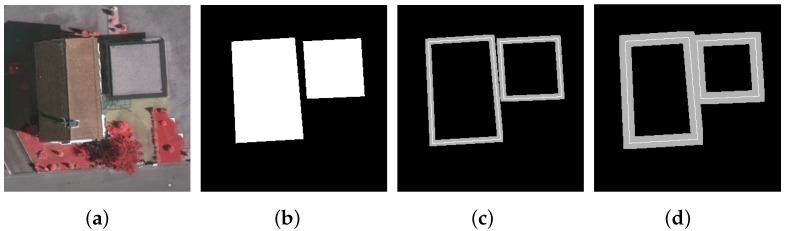
Evaluation of boundary accuracy within trimap regions. (**a**) Test image; (**b**) annotation; (**c**) the trimap with a width of 5 pixels. (**d**) the trimap with a width of 10 pixels. The trimap regions for conducting accuracy evaluation are colored gray and were generated by taking 5 and 10 pixel bands surrounding the object boundaries, respectively.

**Figure 5 sensors-19-01915-f005:**
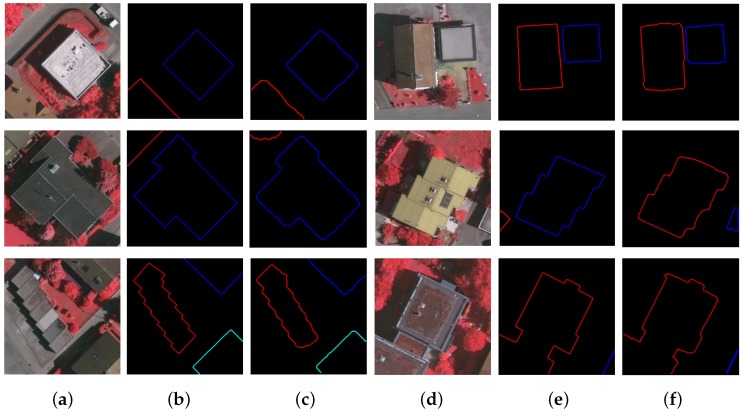
Contour curve extraction on 6 aerial images: (**a**,**d**) test images; (**b**,**e**) ground-truth curves; (**c**,**f**) results generated and used in our corner detection algorithm.

**Figure 6 sensors-19-01915-f006:**
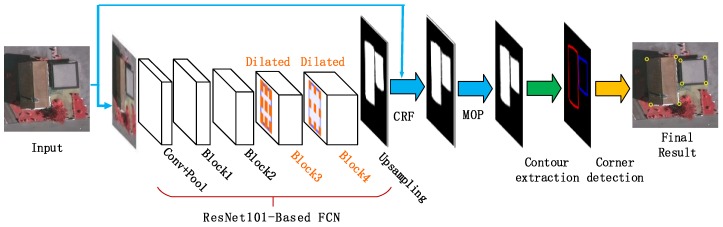
A global view of our proposed building corner detector.

**Figure 7 sensors-19-01915-f007:**
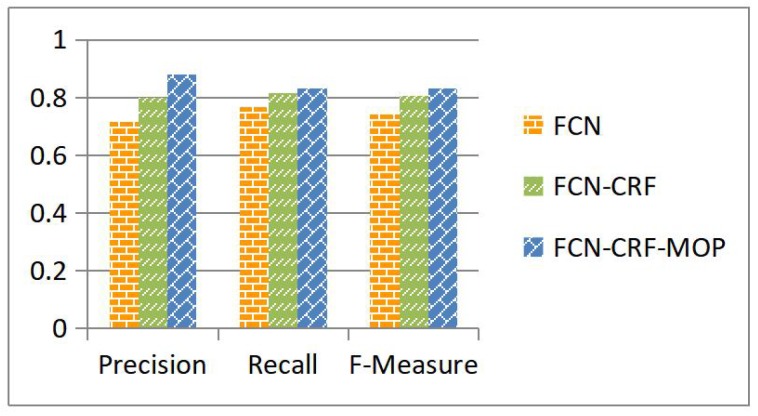
A performance comparison of the existing FCN and FCN-CRF models and our proposed FCN-CRF-MOP model with respect to corner detection.

**Figure 8 sensors-19-01915-f008:**
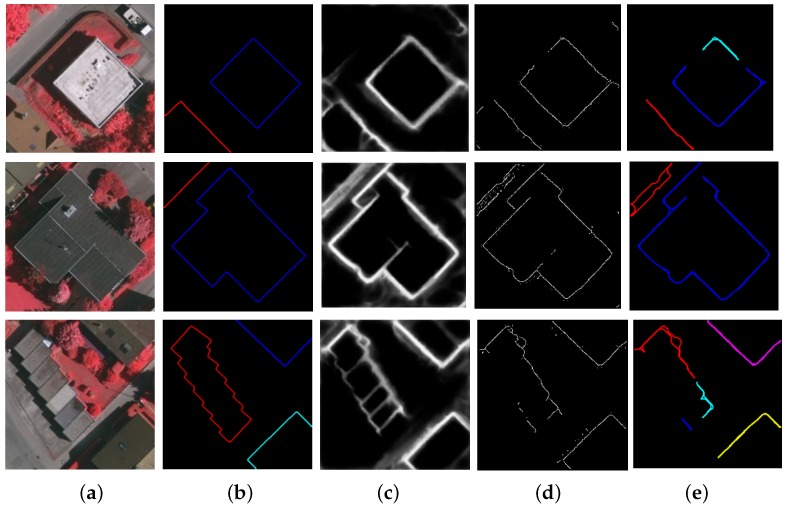
Building contour extraction in aerial images using the Richer Convolutional Features (RCF) [26]. (**a**) Test images; (**b**) ground-truth building contour curves; (**c**) RCF edge probability map; (**d**) RCF binary edge map; (**e**) Final results of the RCF.

**Figure 9 sensors-19-01915-f009:**
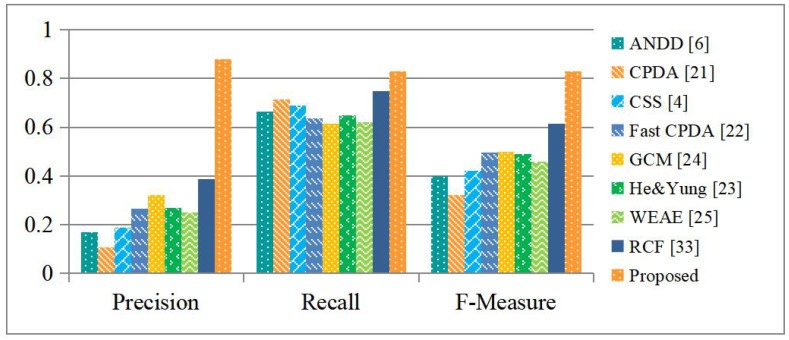
Evaluation results of different corner detectors.

**Figure 10 sensors-19-01915-f010:**
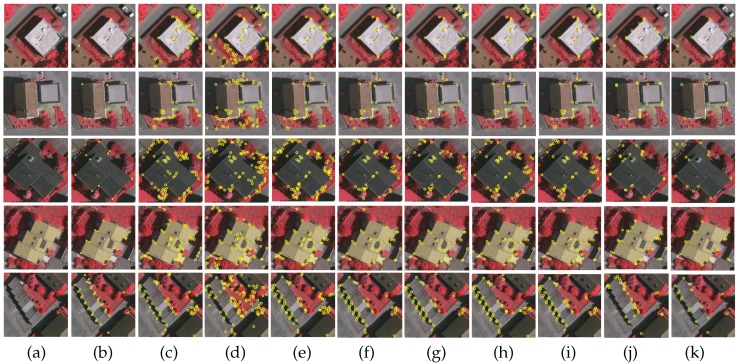
Performance comparison of different corner detectors. (**a**) Test images; (**b**) ground-truth corner points; (**c**) the ANDD [8]; (**d**) the CPDA [27]; (**e**) the CSS [5]; (**f**) the fast CPDA [28]; (**g**) the GCM [30]; (**h**) the He & Yung [29]; (**i**) the WEAE [9]; (**j**) the RCF [26]; (**k**) the proposed model.

**Table 1 sensors-19-01915-t001:** A comparison of different segmentation models under mean IoU (Intersection over Union). The highest result is highlighted in boldface.

Segmentation Method	Mean IoU (%)
FCN	92.83
FCN-CRF	93.16
FCN-CRF-MOP (Proposed)	**93.21**

**Table 2 sensors-19-01915-t002:** Mean IoU of the compared methods within the trimap under varying widths. For each used trimap width, the highest result is highlighted in boldface.

Segmentation Method	Trimap Width (Pixels)					
	5	10	15	20	25	30
FCN	72.59	83.70	88.15	90.50	91.95	92.90
FCN-CRF	74.98	84.91	88.98	91.14	92.46	93.33
FCN-CRF-MOP (Proposed)	**75.12**	**85.00**	**89.05**	**91.50**	**92.50**	**93.36**

**Table 3 sensors-19-01915-t003:** Parameter settings of the compared corner detectors.

Detectors	ANDD	CPDA	CSS	Fast CPDA	GCM	He & Yung	WEAE
η	0.2	0.21	–	0.12	0.0095	1.8	146
$T_{l}$	0.2	0.2	0.2	0.2	0.2	0.2	0.2
$T_{h}$	0.5	0.5	0.5	0.5	0.5	0.5	0.5

**Table 4 sensors-19-01915-t004:** A comparison of different detectors in terms of running time (in seconds).

Detectors	ANDD	CPDA	CSS	Fast CPDA	GCM	He & Yung	WEAE	RCF	Proposed
**Time (s)**	1.03	0.14	0.13	0.04	0.04	0.05	0.06	0.2	0.52

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
