# Peer review of "Building Corner Detection in Aerial Images with Fully Convolutional Networks"

_sensors, 2019, doi:10.3390/s19081915_

Round 1
Reviewer 1 Report
This study aims at detecting building corner in aerial images via adopting fully convolutional networks.
This paper firstly uses FCN to give an initial segmentation of building, and then applies Fully-connected CRF and morphological operation to refine the segmentation results.
Secondly some traditional methods are used for contour extraction. Finally, the simplified evolved method in scale space are used for corner detection. Experimental result validated the proposed method outperform the existed corner detection method.
This study used the complicated FCN for building segmentation. However the final purpose of this study is corner detection, which is implemented by the traditional methods.
I think that the simple way is to apply CNN for direct building curve extraction or corner detection.
As showed in the segmentation results, if only evaluating the tripap with small width, the performance of the segmentation results will be decreased, which manifest the better performance of the segmentation not always generate better result for the final corner detection.
Although the author provided that the post-processing: fully-connected CRF and MOP can improve the performance of trimap evaluation, it seems that no intuitive effort is explored for improving the segmentation results of the building boundary.
The experimental results are compared with the traditional corner detection method, which are not based CNN models.
How about the compared results of the corner detection using the building segmentation with FCN, FCN-CRF, and FCN-CRF-MOP.
Author Response
Response to Reviewer 1 Comments
We would like to thank the reviewers for their valuable time and thoughtful evaluation spent on this manuscript. The paper has been carefully revised according to the reviewers’ comments, and our response to each question is provided below, individually.
This study aims at detecting building corner in aerial images via adopting fully convolutional networks. This paper firstly uses FCN to give an initial segmentation of building, and then applies Fully-connected CRF and morphological operation to refine the segmentation results.
Secondly some traditional methods are used for contour extraction. Finally, the simplified evolved method in scale space are used for corner detection. Experimental result validated the proposed method outperform the existed corner detection method.
Point 1: This study used the complicated FCN for building segmentation. However the final purpose of this study is corner detection, which is implemented by the traditional methods.
I think that the simple way is to apply CNN for direct building curve extraction or corner detection.
Response 1: We agree with the reviewer that it would be simple if CNN could be directly applied for direct building curve extraction or corner detection. However, according to our practice and experience, such approaches are not able to deliver superior performance in building corner detection. In the revised paper, a subsection (Section 4.3) is added to discuss this issue.
For verification, a reference approach is developed in Section 4.3, which directly extracts building contours by using a CNN model trained with the same dataset as our proposed approach. Performance of the reference approach is demonstrated in Figure 8 (Page 10). It can be seen that the extracted building contours could be blurred, fragmented or miss-detected (see the last column of Figure 8). By comparing with the results of our proposed approach (the last column of Figure 5), it can be concluded that the corner detection scheme proposed and used in our work represents a more effective way for detecting building corners from aerial images. Note that the corner detection results produced by using the reference approach have also been presented in Figure 9 and Figure 10, where it is further verified that our proposed approach is more suitable for building corner detection.
Point 2: As showed in the segmentation results, if only evaluating the tripap with small width, the performance of the segmentation results will be decreased, which manifest the better performance of the segmentation not always generate better result for the final corner detection.
Response 2: In the revised paper, Section 4.2 is added to verify the fact that better performance of the segmentation can generate better result for the final corner detection. To be specific, in this added subsection, corner detection is conducted on the segmentation produced with the existing FCN and FCN-CRF models, and our proposed FCN-CRF-MOP model, individually. The results are presented in Figure 7 (Page 9), where it can be seen that our proposed model delivers better results over the existing models in corner detection.
The comment that “if only evaluating the tripap with small width, the performance of the segmentation results will be decreased” is a misunderstanding of what we have expressed, i.e., “the measured accuracy of each method increases as the trimap width increases”. Note that this phenomenon we have expressed is only related to the change of evaluation conditions. In other words, a model could have a low measured accuracy when a stricter evaluation condition is imposed. On the other hand, the model always has “fixed” performance for a given image. To avoid confusion, in the revised paper a description sentence is corrected (Lines 140-141, Page 6): “It is seen that the measured accuracy of each compared method increases as the trimap width increases (that is, when a less-strict evaluation condition is imposed);”
Point 3: Although the author provided that the post-processing: fully-connected CRF and MOP can improve the performance of trimap evaluation, it seems that no intuitive effort is explored for improving the segmentation results of the building boundary.
Response 3: In the paper, Figure 3 demonstrates the progressive improvement achieved by using the CRF and MOP. Since the improvement effect might not be intuitive enough for the readers, we have rewritten the corresponding paragraph for a more specific description (Lines 119-123, Page 5): “ however, the segmentation maps are somewhat blurred and inaccurate. By using the CRF as a post-processing, the boundary edges of buildings become sharper, as shown in the fourth column. Unfortunately, some segmentation flaws, such as spur-like artifacts, are still observed, which could incur many false positives in the subsequent corner detection process.”
In fact, the progressive improvement effect has also be clearly verified with an objective evaluation. Table 1 shows that the FCN has a mean IoU of 92.83%, the FCN-CRF has a mean IoU of 93.16%, and the FCN-CRF-MOP has a mean IoU of 93.21%, repectively.
Point 4: The experimental results are compared with the traditional corner detection method, which are not based CNN models.
Response 4: To our best knowledge, at present there are no CNN models that have been proposed and used for corner detection, either for a generic corner detection task or for a specific corner detection task. In fact, according to our experience, it could be a very hard problem to design and train such a CNN model. Therefore, there are no CNN-based corner detectors that can be compared with our detector.
Point 5: How about the compared results of the corner detection using the building segmentation with FCN, FCN-CRF, and FCN-CRF-MOP.
Response 5: In the revised paper, experiments are conducted to evaluate the corner detection performance of these three models. The results are presented in Figure 7, where it is shown that the F-measure values of corner detection achieved by using the FCN, the FCN-CRF and the FCN-CRF-MOP are 0.74, 0.81 and 0.83, respectively.

Reviewer 2 Report
The paper proposed a method for building corner extraction in high-resolution satellite/aerial images by combinations of building segmentation and contour curve analysis. Although the individual approaches are not very new, the combination is innovative for the task of building corner extraction with the robustness supported by the established methods. I would recommend this for publication in Sensor though I would like to request authors for a revision with some more information and discussions as below.
1. Please highlight some false cases to discuss limitation and possible countermeasures for future works.
2. Abstract and conclusion - Please refer to the accuracy indicators from the results.
3. L85 - Please indicate the spatial resolution of the input data. It is crucial for the parameters in the algorithms that I would request descriptions as the comment below.
3. Section 3.1.2 - Please indicate parameters in the algorithms. As long as I understand correctly, it should be with patch/window size and batch size.
4. L183 - Please indicate parameters in the compared algorithms if applicable.
Author Response
Response to Reviewer 2 Comments
We would like to thank the reviewers for their valuable time and thoughtful evaluation spent on this manuscript. The paper has been carefully revised according to the reviewers’ comments, and our response to each question is provided below, individually.
The paper proposed a method for building corner extraction in high-resolution satellite/aerial images by combinations of building segmentation and contour curve analysis. Although the individual approaches are not very new, the combination is innovative for the task of building corner extraction with the robustness supported by the established methods. I would recommend this for publication in Sensor though I would like to request authors for a revision with some more information and discussions as below.
Point 1: Please highlight some false cases to discuss limitation and possible countermeasures for future works.
Response 1: In the revised paper, Section 4.7 has been added to discuss the above-mentioned issues (Pages 11-12, Lines 275-284). In brief, it is highlighted that for buildings with complex structure, the extracted contours might be blurred and some false negatives could be yielded in corner detection. A possible way to cope with this problem is to add the curvature values of each labeled building contour to the FCNs so that an edge-preserving network model can be trained for building contour extraction. This topic would be one of our future works.
Point 2: Abstract and conclusion - Please refer to the accuracy indicators from the results.
Response 2: Corrections are made as suggested. In the revised paper, it has been clarified both in the abstract and conclusion that the proposed corner detector can achieve an F-measure of 0.83 on the test image set. (Page 1, Line 13; Page 12, Lines 292-293)
Point 3: L85 - Please indicate the spatial resolution of the input data. It is crucial for the parameters in the algorithms that I would request descriptions as the comment below.
Response 3: Corrections are made as suggested. In the revised paper, the spatial resolution of the input data has been specified: “The dataset of aerial images used in the following for training (11,700 images of 321×321 pixels) and testing (450 images of 321 × 321 pixels) is produced from the Vaihingen.” (Page 4, Lines 88-89)
Point 4: Section 3.1.2 - Please indicate parameters in the algorithms. As long as I understand correctly, it should be with patch/window size and batch size.
Response 4: Corrections are made as suggested. In the revised paper, the parameters used in the algorithm have been specified: “The ResNet-101 model contains a 7×7 convolutional layer and 5 residual blocks. Each residual block includes several 3×3 or 1×1 convolutional layer. The latter two blocks are re-purposed by atrous convolution … where the batch size is taken to be 1.” (Page 4, Lines 110-112 and Line 114)
Point 5: L183 - Please indicate parameters in the compared algorithms if applicable.
Response 5: Corrections are made as suggested. In the revised paper, Table 3 is added to summarize the parameter settings of the compared algorithms, and necessary description is also added (Page 10, Lines 244-248).
